# Liraglutide-Induced Hepatotoxicity

**DOI:** 10.3390/biomedicines9020106

**Published:** 2021-01-22

**Authors:** Yaakov Maor, David Ergaz, Stephen D. H. Malnick, Ehud Melzer, Manuela G. Neuman

**Affiliations:** 1Kaplan Medical Center, Institute of Gastroenterology and Hepatology, Hebrew University Medical School of Jerusalem, Rehovot 71600, Israel; yaakovma1@clalit.org.il (Y.M.); EhudM@clalit.org.il (E.M.); 2Internal Medicine Day-Care Kaplan Medical Center, Hebrew University Medical School of Jerusalem, Rehovot 71600, Israel; david.ergaz@clalit.org.il; 3Kaplan Medical Center, Department of Medicine C, Rehovot 71600, Israel; steve@stevemalnickmd.com; 4In Vitro Drug Safety and Biotechnology and Department of Pharmacology and Toxicology, University of Toronto, Toronto, ON M5G 1L5, Canada

**Keywords:** drug-induced liver injury, liraglutide-induced immune hepatitis, lymphocyte toxicity assay, personalized medicine

## Abstract

A 52-year-old woman with a BMI of 31.2 kg/m^2^ was treated with the glucagon-like peptide 1 (GLP-1) agonist liraglutide as part of her weight-reduction program. Following this, she developed an idiosyncratic drug-related liver injury (IDILI). Advances in noninvasive techniques enabled this diagnosis to be established. By employing easily quantifiable methods based on serum biomarkers, we could explore a wide variety of endpoints in assessing personalized DILI. In addition, we can test endpoints that are associated with the drug’s mechanism of action. Personalized medicine and therapeutic pharmacovigilance of incretin-based hypoglycemic agents are needed to ensure the safety of patients.

## 1. Introduction

Liraglutide (Saxenda, Victoza) is a glucagon-like-peptide-1 (GLP-1) analog that acts like incretin to increase insulin secretion through specific beta-cell receptors. Liraglutide shares a 97% homology with human GLP-1 and is approved for glycemic control in adults with type 2 diabetes [1].

The United States Food and Drug Administration (FDA) approved liraglutide as an additional treatment to diet and exercise in order to improve glycemic control in adults with type 2 diabetes mellitus. The drug was not recommended as first-line therapy [1].

Liraglutide has been shown to induce weight reduction in obese individuals [1]. Obesity treatment guidelines recommend the use of therapy in adults who have a body mass index (BMI) equal to 30 kg/m^2^ or higher or in patients with a BMI of 27 kg/m^2^ or higher who have at least one weight-related comorbid condition (e.g., hypertension, dyslipidemia, insulin resistance, type 2 diabetes mellitus).

Glucagon-like peptide 1 receptor agonists (GLP1-RAs) constitute a class of antidiabetic medication. Due to the effects on satiety and weight loss, liraglutide has been repurposed as an anti-obesity medication and is already approved and used in clinical practice.

In 2015, Nuffer and Trujillio published a review specifically based on liraglutide’s role in treating obesity. Data confirmed that liraglutide can achieve a weight loss of 5–10% from baseline. The reported adverse effects were mild gastrointestinal symptoms [2].

The FDA has warned the public about post-marketing reports of acute pancreatitis, including fatal and serious nonfatal cases [3].

Liraglutide improved glycosylated hemoglobin (HbA1c) reductions [4].

The LEADER (Liraglutide Effect and Action in Diabetes: Evaluation of Cardiovascular Outcome Results) clinical trial randomized 9340 participants with type 2 diabetes at high risk for cardiovascular events for a period of between 3.5 to 5 years. The investigators of the trial found an increased risk of acute gallbladder or biliary disease in patients receiving liraglutide (*n* = 141 of 4668) versus placebo (*n* = 88 of 4672) patients [5].

In the PIONEER 4 clinical trial, 14 mg of oral semaglutide was compared with 1.8 mg of liraglutide. The hemoglobin A1c (HbA1c) in patients who had taken liraglutide was reduced by 1.2% compared to 0.9% HbA1c reduction with semaglutide. Adverse events were mild [6,7,8,9].

In a randomized control trial in adolescence with obesity, participants were assigned to the liraglutide and to the placebo group for 56 weeks. The use of liraglutide (3 mg) plus lifestyle therapy led to a significantly greater reduction in the BMI score than placebo plus lifestyle therapy. However, more adolescents in the liraglutide group than in the placebo group had gastrointestinal adverse events (64.8% vs. 36.5%). Additionally, there were serious adverse reactions that led to discontinuation of the trial treatment (10.4% vs. 0%) [10].

To date, there has been only one reported case of liraglutide-induced autoimmune hepatitis [11].

A double-blind placebo-controlled trial showed that participants taking liraglutide 3.0 mg had a statistically significant higher (*p* < 0.0001) weight loss compared with placebo weight loss. The adverse effects were nausea and vomiting [12]. A 16-week randomized, double-blind, placebo-controlled pilot trial studying the effects of liraglutide (3.0 mg) versus placebo in obese individuals described nausea as a side-effect in 12 out of 19 patients treated with liraglutide compared with 4 of the 21 in the placebo group [13].

We hereby report a patient with reversible liraglutide-induced hepatotoxicity. Our aim was to confirm the role of this drug in the hepatotoxic reaction of the patient and to elucidate the mechanisms of liraglutide-induced liver injury.

## 2. Case Report

A 52-year-old obese Caucasian (BMI 31.2 kg/m^2^) woman was treated for weight control with subcutaneous liraglutide (Saxenda, Novo Nordisk, Bagsvaerd, Denmark). The initial dose of 0.6 mg daily was increased gradually to 3 mg daily. After three months, the patient lost 10 kg of weight, and the BMI decreased to 27.5 kg/m^2^.

The patient was not aware of any liver disease. She consulted the clinicians due to increased liver enzyme levels. The biochemical picture showed a mixed cholestatic-hepatocellular pattern.

Alanine aminotransferase (ALT) was increased to 547 IU/L, aspartate aminotransferase (AST) to 268 IU/L, alkaline phosphatase (ALP) to 390 IU/L, gamma-glutamyl transferase (γGT) to 427 IU/L (Figure 1) and total bilirubin levels to 1.3 mg/dL (Figure 2). In addition, there was mild eosinophilia of 8.5% with an absolute count of 600/uL and the erythrocyte sedimentation rate (ESR) was elevated to 59 mm/h. Anti body testing revealed only anti-histone antibodies positive at 52 IU/mL. Anti-mitochondrial antibody (AMA) and anti-smooth muscle actin antibody (ASMA) were negative. The viral analyses for hepatitis B (HBV) and C (HCV), Epstein–Barr virus (EBV), cytomegalovirus (CMV) as well as herpes simplex virus 6 (HHV6) analyses were negative. Liver genetic diseases were also excluded.

Abdominal ultrasound showed a fatty liver. The calculated Roussel Uclaf Causality Assessment Method (RUCAM) score was 7. The patient did not smoke or misuse alcohol or other substances. A lymphocyte toxicity assay (LTA) detected liraglutide-induced toxicity of 35% (normal 10–15%). The LTA used the patient lymphocyte population as an organ target for the toxic effect of drug metabolite(s). The LTA clearly demonstrated that this drug produced the toxic damage.

### Clinical Management

Liraglutide was discontinued with gradual clinical improvement and normalization of her liver enzymes over a three-month period.

## 3. Materials and Methods

During the hospitalization period, the routine analyses (liver enzymes, bilirubin, hematology and virology) were performed at the Kaplan Medical Center. The additional special analysis (apoptosis, LTA, cytokines, chemokines, Epstein–Bar virus (EBV) and cytomegalovirus (CMV)) were performed at the In Vitro Drug Safety and Biotechnology lab.

### 3.1. LTA

The LTA is based on the mitochondrial succinate dehydrogenase (SDH) activity to confirm sensitivity/toxicity reactions to the investigated therapy [14]. The LTA is a non-invasive, precise and personal test used to determine causality in drug-induced hypersensitivity [14].

Lymphocytes are easily accessible and contain the enzymes required for detoxification and the individual’s genetic deficiencies. The LTA consists of incubating patient lymphocytes with the drug in the presence or absence of murine hepatic microsomes. The percentage of cell death is calculated by the following formula: cells treated with the drug in the presence of microsomes—cells treated only with drug/100 [14]. In vitro toxicity has been assessed by using MTT {3-(4,5-dimethylthiazol-2-yl-2,5-diphenyl tetrazolium bromide)} [14]. The sample was studied in triplicate, and the mean percent of dead lymphocytes was determined.

### 3.2. Cytokines

The pro- and anti-inflammatory cytokines and adipokines were measured in the sera and in the media using available cytokine ELISA kits. Our laboratory uses special in-house validated controls.

The cytokines were quantitatively determined in serum and media, using enzyme-linked immunosorbent assay (ELISA). Cytokine Kits were as follows: TGFβ (R&D Systems, Inc., Minneapolis, MN, USA), IL-1, VEGF, TNF-α levels, IL-6, IL-8, RANKL, RANTES (pg/mL), (PeproTech Asia, Rehovot, Israel), NFκB (InVitrogen, Termo Fisher, Waltham, MA, USA). The tests were performed according to manufacturer specifications. For cytokine determination, each specimen was analyzed in duplicate with 95% sensitivity and 92% specificity. Our measurement system demonstrates strong correlations across replicates with correlation coefficients > 0.99, ensuring reliable detection of differences in cytokine levels between biological samples.

#### Apoptosis

Liver cell apoptosis is triggered by host conditions and the presence of toxins. To confirm the mechanism, we chose to analyze multiple apopto-necrotic markers.

Cytokeratin 18 encodes the type I, chain keratin 18. Keratin 18, together with its keratin 8, is expressed in single-layer epithelial tissues of the body. Mutations in this gene have been linked to cryptogenic cirrhosis. We measured the cytokeratins in sera and tissue media using mitochondrial markers: M30 and M65. M30 is specific for apoptosis, and M65 combines death processes from both apoptosis and necrosis. The M30 Apoptosense^®^ ELISA measures the levels of soluble caspase-cleaved K18 (ccK18) fragments containing the K18Asp396 neo-epitope. The ccK18 level increases during apoptosis and is inhibited by the caspase-inhibitor zVAD-fmk M65^®^. The marker-cleaved cytokeratins CK 18 and CK 8 (M30 and M65) were quantified using kits from Bender MedSystems (Vienna, Austria). We used the same procedures described by us in previous studies.

The correlation coefficient was linear (*r* = 0.990). NFκB is predominantly localized in the cytoplasm with a double role as a mediator of cell survival as well as an up-regulator of Fas-FasL apoptosis. The standards were from the National Institute for Biological Standards and Controls (NIBSC, Herts, UK). These methods are standardized in our laboratory according to the procedures described [14].

## 4. Results

### 4.1. Immuno-Mechanism

The lymphocyte toxicity assay demonstrated a liraglutide-induced toxic response to the patient’s cells: 34% toxicity. The ten control individuals that took the same drug and did not present an adverse event had 0–15% toxicity.

In Table 1, we present the levels of cytokines, chemokines, apoptotic and necrotic markers in sera of the patients and controls. In Table 2, we show the values of the same parameters in the media in which the lymphocytes were exposed to the drug in the presence of microsomes for 24 h. The mitochondrial markers revealed a high necrotic score. The pro-inflammatory cytokines (×time control levels) in serum were as follows: TNF (tumor necrosis factor alpha) × 26 upper limit of normal (ULN); interlekin (IL)1 beta × 5 ULN; IL6 × 3 ULN; and IL8 × 6 ULN. Vascular endothelial growth factor (VEGF) was 3 × ULN. Adipokine levels were leptin × 6 ULN, adiponectin × 0.5 ULN and gherelin × 2 ULN. The dysregulation of nuclear factor KB (NFKB) signal transduction pathway × 3 ULN in our patient can be associated with inflammatory or immune diseases.

### 4.2. Kinetics of Serum Enzymes and Bilirubin

Figure 1 presents the kinetics of AST, ALT, ALP and γGT from the moment that the patient was hospitalized, several points after and 11 months after hospitalization.

Figure 2 presents the kinetics of bilirubin from the moment that the patient was hospitalized, when she was released and three times after hospitalization.

### 4.3. Cytokines and Chemokines

The exposure of the patient’s lymphocytes to the drug in the presence of microsomes increased the production of pro-inflammatory mediators. These mediators were released into the media during the 24 h of exposure.

## 5. Discussion

The imbalance between the pro- and anti-inflammatory cytokines leads to inflammation and cell death via necrosis. DILI can be classified as either a direct-intrinsic liver acute injury or an acute liver injury due to idiosyncratic toxicity (IDILI). In this case, we believe that the adverse reaction is idiosyncratic. The eosinophil count was high. The LTA clearly demonstrates that the drug is responsible for the toxic event. Moreover, the drug metabolite(s) is/are responsible for a high inflammatory process as demonstrated by cytokine and chemokine levels.

This spectrum of drug-induced liver hepatotoxicity encompasses a direct-intrinsic or indirect toxicity produced by a drug or a toxicant [15,16,17,18,19]. Diagnosis of the liver injury requires clear differentiation of the disease stages. Causality for the present case was established by employing the liver-specific causality assessment method RUCAM (Roussel UCLAF Causality Assessment Method) [19,20,21].

Based on the physiologic property of GLP-1, a study on patients with non-alcoholic steatohepatitis (NASH) was conducted. The impact of liraglutide on hepatic histology was first assessed in the Liraglutide Effect and Action in Diabetes LEAD trial that compared forty-eight weeks of subcutaneous liraglutide (1.8 mg/day) to placebo in 52 patients with biopsy-proven NASH. The primary endpoint was the resolution of NASH without impairment of fibrosis. Liraglutide significantly improved steatosis and hepatocyte ballooning, but no significant differences were seen in lobular inflammation and in the non alcoholic fatty liver disease (NAFLD) activity score [22,23,24,25,26].

The patient was diagnosed with NAFLD only upon her hospitalization. The diagnosis was based on the finding of a hyperechogenic liver in an obese patient with diabetes and hyperlipidemia. There was no evidence for other causes of steatosis. In addition, she had anti-histone antibody positivity.

Magorivska et al. analyzed the anti-histone IgG in patients with systemic lupus erythematosus (SLE) [27]. The authors demonstrated that the sialylated anti-histone IgG altered the cytokine secretion profiles from IL-6/IL-8 to TNF-α/IL-1β thereby regulating chronic inflammation [28]. We suggest that the hepatotoxicity demonstrated in our patient may be the result of an incretin analog adverse reaction with a long latency period. This case should raise awareness about the potential adverse effects of GLP-1 agonists.

## 6. Conclusions

The case presents a young obese individual treated with the long-acting GLP-1 receptor agonist liraglutide. She achieved significant weight loss but developed liraglutide-induced DILI via an inflammatory response.

## Figures and Tables

**Figure 1 biomedicines-09-00106-f001:**
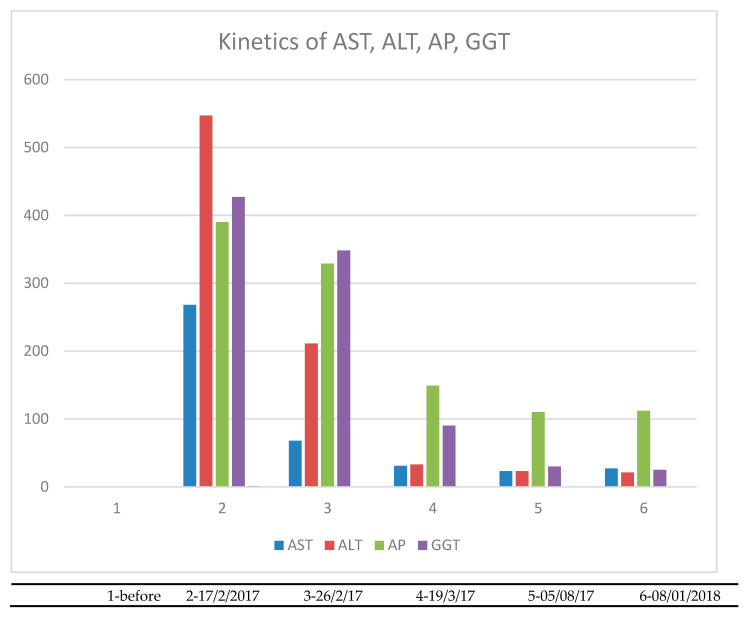
Kinetics of liver enzymes during the hospitalization and observation.

**Figure 2 biomedicines-09-00106-f002:**
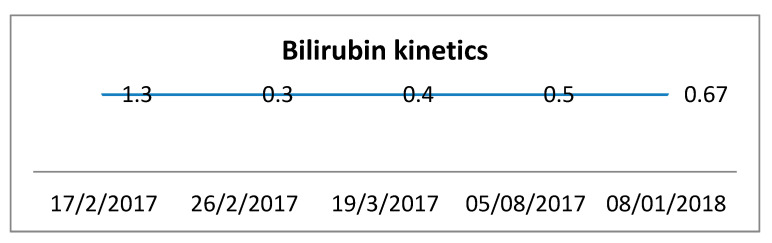
Bilirubin kinetics from the moment that the patient was hospitalized, several points after and 11 months after hospitalization.

**Table 1 biomedicines-09-00106-t001:** Serum cytokine levels.

Cytokines	Normal Values	Patient
IL-1 pg/mL	24–50	253
IL-6 pg/mL	30–60	176
IL-8 pg/mL	20–60	357
NFκB pg/mL	15–40	123
RANTES pg/mL	15–50	215
TNF-α pg/mL	25–50	1350
TGF-β ng/mL	20–40	68
VEGF pg/mL	25–60	183
sRANKL pg/mL	60–80	74
ccK18 (M-30) U/L	68–132	140
ccK8 (M65) U/L	62–213	322

Normal values: levels of the same elements in healthy controls (500); ccK: cytokeratin; mL: milliliter; ng: nanogram; pg: picogram; L: liter; U: units; IL: interleukin; NFκB: nuclear factor kappa B; RANKL: receptor activator of nuclear factor-κB ligand; RANTES: regulated upon activation, normal T cell expressed secreted; TGF: transforming growth factor; TNF: tumor necrosis factor; VEGF: vascular endothelial growth factor.

**Table 2 biomedicines-09-00106-t002:** Levels of cytokines, chemokines, apoptosis and necrosis markers released by lymphocytes exposed to liraglutide in the media.

Cytokines	Normal Values	Patient
IL-1 pg/mL	24–50	983
IL-6 pg/mL	30–60	30
IL-8 pg/mL	20–60	517
NFκB pg/mL	15–40	416
sRANKL pg/mL	60–80	35
RANTES pg/mL	15–50	215
TNF-α pg/mL	25–50	590
TGF-β ng/mL	20–40	56
VEGF pg/mL	25–60	135
ccK18 (M-30) U/L	68–132	110
ccK8(M65) U/L	62–213	318

Normal values refer to the levels of elements measured in the media in which we incubated the lymphocytes of individuals not sensitive to liraglutide. The lymphocytes were exposed to the drug in the presence of microsomes.

## Data Availability

All the authors had full access to all the data in the study and took responsibility for the integrity of the data and the accuracy of the data analysis. All the authors contributed to the study concept and design: acquisition, analysis and interpretation of data, drafting of the manuscript and revision of the manuscript for intellectual content. This research received no external funding. The laboratory analysis and medical support were part of the standard of care offered by the hospital. In Vitro Drug Safety and Biotechnology provided technical and material support for special analyses. Dr. Neuman is the founder of In Vitro Drug Safety and Biotechnology. The funders had no role in the design and conduct of the study; collection, management, analysis and interpretation of the data; preparation, review or decision to submit the manuscript for publication. Reported results can be found in the hospital recording database.

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
