# Peer review of "Liraglutide-Induced Hepatotoxicity"

_biomedicines, 2021, doi:10.3390/biomedicines9020106_

Round 1

Reviewer 1 Report

just a few brief considerations, on a work that I consider important since it indicates the need for a less invasive but very useful investigative procedure in order to highlight any rare but serious drug-induced toxicity.

This case is the 2nd report of liraglutide-induced - hepatotoxicity.

The search for serum biomarkers as indicators of effects of the liraglutide, may be a very useful noninvasive strategy for monitoring adverse effects of liraglutide, and more generally of the class of glucagon-like peptide 1 agonists.  

recommend some references for this manuscript:

1b) Arne Astrup, Stephan Rössner, Luc Van Gaal, Aila Rissanen, Leo Niskanen, Mazin Al Hakim, Jesper Madsen, Mads F Rasmussen, Michael E J Lean,

on behalf of the NN8022-1807 Study Group*

Effects of liraglutide in the treatment of obesity: a randomised, double-blind, placebo-controlled study.

Lancet 2009; 374: 1606–16

1c) Houssam Halawi*, Disha Khemani*, Deborah Eckert, Jessica O’Neill, Hoda Kadouh, Karen Grothe, Matthew M Clark, Duane D Burton, Adrian Vella,

Andres Acosta, Alan R Zinsmeister, Michael Camilleri.

Effects of liraglutide on weight, satiation, and gastric functions in obesity: a randomised, placebo-controlled pilot trial

Lancet Gastroenterol Hepatol, 2017; 2: 890–99

Author Response

Thank you very much for the reviewing our manuscript and for proposing the references. We introduced the references (13,14).

Also many thanks for pointing that it is the secon case of hepatotoxicity. We made the correction.

Reviewer 2 Report

In the submitted manuscript entitled, “Liraglutide-induced hepatotoxicity,” the authors present a case of likely drug-induced liver injury (DILI) related to liraglutide along with in vitro data suggesting an immune mechanism of injury. The manuscript is concise and generally is clearly written. Additionally, the topic is important as GLP-1 agonists are likely to be increasingly used in patients with or at risk for chronic liver disease. However, there are a number of concerns that need to be addressed.

Major Comments:

  1. As a case report, there are numerous pieces of important clinical data that are missing, which include but are not limited to: liver enzymes and/or liver imaging at baseline, if available; trend of liver enzymes, including after drug discontinuation; data on comorbidities, if any; full medication list, or an indication that the patient was not taking other medications; physical exam findings, including presence or absence of fever, rash, stigmata of chronic liver disease, etc; liver biopsy data, if any; full results of autoantibody testing (ANA, anti-mitochondrial antibody, anti-smooth muscle actin antibody, etc; anti-histone antibody is of unknown significance in liver injury).
  2. Related to comment #1, it is not clear to this reviewer whether the patient had underlying liver disease (NAFLD or other) based on the information presented in the manuscript. On the fifth page of the text, it states, “…woman was evaluated for increased liver enzyme levels with a mixed cholestatic-hepatocellular pattern. She was recommended subcutaneous liraglutide.” If her liver enzymes were elevated at baseline, what work-up was done at that time, if any? On the second to last text page, it is stated that DILI may occur “…due to an idiosyncratic toxicity confined to cases like our young woman without preexisting liver disease.” Later, the conclusion states, “This case presents a young obese individual with non-alcoholic fatty liver disease…”
  3. The abstract states that the patient “…developed autoimmune hepatitis associated with the therapeutic intervention.” Autoimmune hepatitis is a distinct clinical entity with specific diagnostic criteria and histologic findings, and this reviewer could not identify evidence in the manuscript that the patient had developed autoimmune hepatitis.
  4. The abstract states, “…we can test endpoints that are associated with the drug’s mechanism of action.” This reviewer was not able to identify a direct connection between this statement and the in vitro data presented.
  5. The in vitro data should be more clearly presented. For example, was the lymphocyte toxicity assay carried out in the absence of microsomes as well (i.e., just lymphocytes and drug)? This is an important control. Additionally, the data presented in Table 2 are confusing: were these data obtained in the presence of microsomes as well? Were control lymphocytes from other patient(s) used in this assay? It does not seem that the normal values reported in the table were derived from side-by-side controls, since they appear to be the same data ranges included as normal values in Table 1.
  6. On the fifth text page, it is stated, “To date there have been two reported cases of liraglutide-induced autoimmune hepatitis.” The first of the cited articles reported a case of DILI (not autoimmune hepatitis) from a different medication, sitagliptin.

Minor Comments:

  1. In this reviewer’s view, it is not necessary to dedicate so much attention and space to reviewing published data from clinical trials of GLP-1 agonists, particularly the data related to efficacy rather than toxicity.
  2. Table 2 – the legend indicates that patient lymphocytes were incubated with denosumab, whereas it should presumably say “liraglutide.”
  3. There are a few other, more minor, typographical errors or missing/duplicated terms in the manuscript.

Author Response

Thank you very much for accepting to review our manuscript.

I will respond to each point in italic

Major Comments:

  1. As a case report, there are numerous pieces of important clinical data that are missing, which include but are not limited to: liver enzymes and/or liver imaging at baseline, if available; trend of liver enzymes, including after drug discontinuation; data on comorbidities, if any; full medication list, or an indication that the patient was not taking other medications; physical exam findings, including presence or absence of fever, rash, stigmata of chronic liver disease, etc; liver biopsy data, if any; full results of autoantibody testing (ANA, anti-mitochondrial antibody, anti-smooth muscle actin antibody, etc; anti-histone antibody is of unknown significance in liver injury).

 Thank you for the reviewer we introduced the requested data,

It is true that there is no reported data between the link antihistone antibody to liver damage. The link between high levels of the marker in a auto-immune disease (rheumatoid arthritis) might suggest a possible immune reaction in the present case.

  1. Related to comment #1, it is not clear to this reviewer whether the patient had underlying liver disease (or other) based on the information presented in the manuscript. On the fifth page of the text, it states, “…woman was evaluated for increased liver enzyme levels with a mixed cholestatic-hepatocellular pattern. She was recommended subcutaneous liraglutide.”

Thank you for the observation. We explained the idea better in the text. The person received Liraglutide for her obesity. She was not aware for NAFL or any other liver problem, previously. She consulted the physicians only when she had the liver -related symptoms.

On the second to last text page, it is stated that DILI may occur “…due to an idiosyncratic toxicity confined to cases like our young woman without preexisting liver disease.” Later, the conclusion states, “This case presents a young obese individual with non-alcoholic fatty liver disease…”

The diagnoses of NAFLD was made by the two hepatologists (Y. Maor and S. Malnick) only when the patient was hospitalized not previously. We deleted the NAFL from conclusion.

  1. The abstract states that the patient “…developed autoimmune hepatitis associated with the therapeutic intervention.” Autoimmune hepatitis is a distinct clinical entity with specific diagnostic criteria and histologic findings, and this reviewer could not identify evidence in the manuscript that the patient had developed autoimmune hepatitis.

We made a change in the abstract 'she developed and idiosyncratic drug-related liver injury (IDILI) with immune hepatitis ".

  1. The abstract states, “…we can test endpoints that are associated with the drug’s mechanism of action.” This reviewer was not able to identify a direct connection between this statement and the in vitro data presented.

I showed the connection. LTA  demonstrated liraglutide -induced toxicity. The LTA showed a 35% toxicity.

  1. The in vitro data should be more clearly presented. For example, was the lymphocyte toxicity assay carried out in the absence of microsomes as well (i.e., just lymphocytes and drug)? This is an important control.

Thank you fro giving the opportunity to explain the test.

The test is carried out both in the lymphocytes + drug and the lymphocytes + drug + microsomes and we explain the procedure and the calculation. Certainly the presence of microsomes leads to the formation of toxic metabolites. These metabolites are responsible for toxic effect of the drug on cells.

Additionally, the data presented in Table 2 are confusing: were these data obtained in the presence of microsomes as well? Were control lymphocytes from other patient(s) used in this assay? It does not seem that the normal values reported in the table were derived from side-by-side controls, since they appear to be the same data ranges included as normal values in Table 1.

There were not side by side controls. The normal controls are derived from  individuals exposed to the same drug and did not present adverse reactions to the drug. The exposure and analysis was performed at different points of time in my lab. I also introduced the explanation in the text.

  1. On the fifth text page, it is stated, “To date there have been two reported cases of liraglutide-induced autoimmune hepatitis.” The first of the cited articles reported a case of DILI (not autoimmune hepatitis) from a different medication, sitagliptin.

Thank you very much for the observation. We changed the phrase to reflect the truth.

Minor Comments:

  1. In this reviewer’s view, it is not necessary to dedicate so much attention and space to reviewing published data from clinical trials of GLP-1 agonists, particularly the data related to efficacy rather than toxicity.

 We reduced it.

  1. Table 2 – the legend indicates that patient lymphocytes were incubated with denosumab, whereas it should presumably say “liraglutide.”

Thank you for the observation, we corrected it.

  1. There are a few other, more minor, typographical errors or missing/duplicated terms in the manuscript.

We corrected the mistakes

Reviewer 3 Report

I read with interest the manuscript within form of a case report “Liraglutide-induced – hepatotoxicity”

This article is divided into three parts:

(1)  in the introduction section, the authors describe the drug (indication, efficacy, safety …) and review the main studies about Liraglutide and others GLP1-RA class. This part is too long (with redundant sentences) and irrelevant. An effort to shorten this paragraph would be appreciate.

(2) The case report is clear and well written. The authors describe at the end of the 4th month a clinical picture of myocardial infarction without any further precision. What is the impact of this pathology ?

(3) the methods section is long and the description of the methodology is too complex for the initiated and probably too simple for others. Please shorten this part.

Finally, I’m not immunologist but the authors must convince the readers of the link between inflammation and autoimmunity in this case. Didn't the patient have autoimmunity before the start of treatment?  

In the manuscript, no proposal for the management of this monitoring of this treatment is made.

Author Response

I read with interest the manuscript within form of a case report “Liraglutide-induced – hepatotoxicity”

This article is divided into three parts:

(1)  in the introduction section, the authors describe the drug (indication, efficacy, safety …) and review the main studies about Liraglutide and others GLP1-RA class. This part is too long (with redundant sentences) and irrelevant. An effort to shorten this paragraph would be appreciate.

We shorten the introduction.

(2) The case report is clear and well written. The authors describe at the end of the 4th month a clinical picture of myocardial infarction without any further precision. What is the impact of this pathology ?

(3) the methods section is long and the description of the methodology is too complex for the initiated and probably too simple for others. Please shorten this part.

We reduced the methodology part.

Finally, I’m not immunologist but the authors must convince the readers of the link between inflammation and autoimmunity in this case. Didn't the patient have autoimmunity before the start of treatment?  

The patient was not known to have autoimmunity. The histone autoantibodies vote for a possible immune problem.

In the manuscript, no proposal for the management of this monitoring of this treatment is made.

The management of the patient was introduced.

Round 2

Reviewer 2 Report

In the revised manuscript entitled, “Liraglutide-induced hepatotoxicity,” the authors have improved their description of a case of likely drug-induced liver injury (DILI) related to liraglutide along with the presentation of in vitro data supporting an immune mechanism of injury. The manuscript is concise, and the presentation is improved from initial submission. A few questions and/or points of clarification are listed below.

Major Comments:

  1. The term “autoimmune hepatitis” has appropriately been removed from the manuscript, but in the abstract, this term has been replaced by “immune hepatitis.” I am not aware of the clinical meaning of this term, and I fear that readers will interpret this as implying “autoimmune hepatitis,” which is not accurate. This type of idiosyncratic DILI is generally postulated to have an immune component to the mechanism of injury, as supported by the in vitro data for this patient, so in this reviewer’s opinion, it is simpler to just say that rather than to introduce a term such as “immune hepatitis,” which is confusing.
  2. Is it correct that the laboratory-determined normal values for the tested cytokines are the same in patient serum (Table 1) as for medium in which lymphocytes were incubated with liraglutide (Table 2)? I am surprised that the normal ranges are identical in the two settings.
  3. On page 4 it is stated, “The biopsy and the blood of the patient was taken during her hospitalization and analyzed in the laboratory.” Did the patient have a liver biopsy? If so, her liver histology results would enrich the case report. If not, this statement should be clarified.

Minor Comments:

  1. In this reviewer’s view, the attention and space given to reviewing published data from clinical trials of GLP-1 agonists could still be reduced.
  2. It would be helpful to refer directly to Tables 1 and 2 in the text so that the reader understands exactly which data are being referenced when reading the Results section.
  3. There are still minor typographical errors or missing/duplicated words in the manuscript that will need to be fixed in the copy-editing process.

Author Response

Thank you for the suggestion. you will find my answers in italic.

Major Comments:

  1. The term “autoimmune hepatitis” has appropriately been removed from the manuscript, but in the abstract, this term has been replaced by “immune hepatitis.” I am not aware of the clinical meaning of this term, and I fear that readers will interpret this as implying “autoimmune hepatitis,” which is not accurate.This type of idiosyncratic DILI is generally postulated to have an immune component to the mechanism of injury, as supported by the in vitro data for this patient, so in this reviewer’s opinion, it is simpler to just say that rather than to introduce a term such as “immune hepatitis,” which is confusing.

 We changed the words in the abstract.

  1. Is it correct that the laboratory-determined normal values for the tested cytokines are the same in patient serum (Table 1) as for medium in which lymphocytes were incubated with liraglutide (Table 2)? I am surprised that the normal ranges are identical in the two settings.

On page 4 it is stated,

  1. “The biopsy and the blood of the patient was taken during her hospitalization and analyzed in the laboratory.” Did the patient have a liver biopsy? If so, her liver histology results would enrich the case report. If not, this statement should be clarified.

The patient did not have a liver biopsy. The word "biopsy" was erased

Minor Comments:

  1. In this reviewer’s view, the attention and space given to reviewing published data from clinical trials of GLP-1 agonists could still be reduced.
  2. It would be helpful to refer directly to Tables 1 and 2 in the text so that the reader understands exactly which data are being referenced when reading the

Results section.

  1. There are still minor typographical errors or missing/duplicated words in the manuscript that will need to be fixed in the copy-editing process.

Thank you for the helpful remarks.

Reviewer 3 Report

The authors have improved their manuscript and in particular the discussion and conclusion that is more appropriate to understand the link between Liraglutide and hepatotoxicity. I would say that the questions raised are answered quite fairly.

There are some typographical errors; case repor(t), the figure can be improved on the presentation.  

The introduction section is too long for a case report. The studies on Liraglutide and others GLP1-RA class are not useful.

Please delete these sentences: “After 4 months of therapy she complained of atypical chest pain. Stress and stress echocardiography revealed ischemia of the inferior cardiac wall. Routine laboratory results” no impact on the case report and cause confusion.

Author Response

Thank you for the suggestion. you will find my answers in italic.

The authors have improved their manuscript and in particular the discussion and conclusion that is more appropriate to understand the link between Liraglutide and hepatotoxicity. I would say that the questions raised are answered quite fairly.

There are some typographical errors; case repor(t), the figure can be improved on the presentation.  

The introduction section is too long for a case report. The studies on Liraglutide and others GLP1-RA class are not useful.

Thank you for the very useful suggestions.

Please delete these sentences: “After 4 months of therapy she complained of atypical chest pain. Stress and stress echocardiography revealed ischemia of the inferior cardiac wall. Routine laboratory results” no impact on the case report and cause confusion.

Thank you for the very useful review, we made the requested change.